# BLOCK SKIM TRANSFORMER
# FOR EFFICIENT QUESTION ANSWERING

## ABSTRACT

Transformer-based encoder models have achieved promising results on natural language processing (NLP) tasks including question answering (QA). Different from sequence classification or language modeling tasks, hidden states at all positions are used for the final classification in QA. However, we do not always need all the context to answer the raised question. Following this idea, we proposed Block Skim Transformer (BST ) to improve and accelerate the processing of transformer QA models. The key idea of BST is to identify the context that must be further processed and the blocks that could be safely discarded early on during inference. Critically, we learn such information from self-attention weights. As a result, the model hidden states are pruned at the sequence dimension, achieving significant inference speedup. We also show that such extra training optimization objection also improves model accuracy. As a plugin to the transformer based QA models, BST is compatible to other model compression methods without changing existing network architectures. BST improves QA models' accuracies on different datasets and achieves $1.6\times$ speedup on BERT$_{\text{large}}$ model.

## 1 INTRODUCTION

With the rapid development of neural networks in NLP tasks, the Transformer (Vaswani et al., 2017) that uses multi-head attention (MHA) mechanism is a recent huge leap (Goldberg, 2016). It has become a standard building block of recent NLP models. The Transformer-based BERT (Devlin et al., 2018) model further advances the model accuracy by introducing self-supervised pre-training and has reached the state-of-the-art accuracy on many NLP tasks.

One of the most challenging tasks in NLP is question answering (QA) (Huang et al., 2020). Our key insight is that when human beings are answering a question with a passage as a context, they do *not* spend the same level of comprehension for each of the sentences equally across the paragraph. Most of the contents are quickly skimmed over with little attention on it. However, in the Transformer architecture, all tokens go through the same amount of computation, which suggests that we can take advantage of that by discarding many of the tokens in the early layers of the Transformer. This redundant nature of the transformer induces high execution overhead on the input sequence dimension.

To mitigate the inefficiencies in QA tasks, we propose to assign more attention to some blocks that are more likely to contain actual answer while terminating other blocks early during inference. By doing so, we reduce the overhead of processing irrelevant texts and accelerate the model inference. Meanwhile, by feeding the attention mechanism with the knowledge of the answer position directly during training, the attention mechanism and QA model's accuracy are improved.

In this paper, we provide the first empirical study on attention featuremap to show that an attention map could carry enough information to locate the answer scope. We then propose Block Skim Transformer (BST), a plug-and-play module to the transformer-based models, to accelerate transformer-based models on QA tasks. By handling the attention weight matrices as feature maps, the CNN-based Block Skim module extracts information from the attention mechanism to make a skim decision. With the predicted block mask, BST skips irrelevant context blocks, which do not enter subsequent layers' computation. Besides, we devise a new training paradigm that jointly trains the Block Skim

objective with the native QA objective, where extra optimization signals regarding the question position are given to the attention mechanism directly.

In our evaluation, we show BST improves the QA accuracy and F1 score on all the datasets and models we evaluated. Specifically, $BERT_{large}$ is accelerated for $1.6\times$ without any accuracy loss and nearly $1.8\times$ with less than 0.5% F1 score degradation.

This paper contributes to the following 3 aspects.

- We for the first time show that an attention map is effective for locating the answer position in the input sequence.

- We propose Block Skim Transformer (BST), which leverages the attention mechanism to improve and accelerate transformer models on QA tasks. The key is to extract information from the attention mechanism during processing and intelligently predict what blocks to skim.

- We evaluate BST on several Transformer-based model architectures and QA datasets and demonstrate BST 's efficiency and generality.

## 2 RELATED WORK

**Recurrent Models with Skimming.** The idea to skip or skim irrelevant section or tokens of input sequence has been studied in NLP models, especially recurrent neural networks (RNN) (Rumelhart et al., 1986) and long short-term memory network (LSTM) (Hochreiter & Schmidhuber, 1997). LSTM-Jump (Yu et al., 2017) uses the policy-gradient reinforcement learning method to train a LSTM model that decides how many time steps to jump at each state. They also use hyper-parameters to control the tokens before jump, maximum tokens to jump, and maximum number of jumping. Skim-RNN (Seo et al., 2018) dynamically decides the dimensionality and RNN model size to be used at next time step. In specific, they adopt two "big" and "small" RNN models and select the "small" one for skimming. Structural-Jump-LSTM (Hansen et al., 2018) use two agents to decide whether jump a small step to next token or structurally to next punctuation. Skip-RNN (Campos et al., 2017) learns to skip state updates thus results in reduced computation graph size. The difference of BST to these works are two-fold. Firstly, the previous works make skimming decisions based on the hidden states or embeddings during processing. However, we are the first to analyze and utilize the attention relationship for skimming. Secondly, our work is based on Transformer model (Vaswani et al., 2017), which has outperformed the recurrent type models on most NLP tasks.

**Transformer with Input Reduction.** On contrast to aforementioned recurrent models, in the processing of Transformer-based model, all input sequence tokens are calculated in parallel. As such, skimming can be regarded as reduction on sequence dimension. PoWER-BERT (Goyal et al., 2020) extracts input sequence token-wise during processing based on attention scores to each token. During the fine-tuning process for downstream tasks, Goyal et al. proposes soft-extract layer to train the model jointly. Funnel-Transformer (Dai et al., 2020) proposes a novel pyramid architecture with input sequence length dimension reduced gradually regardless of semantic clues. For tasks requiring full sequence length output, like masked language modeling and extractive question answering, Funnel-Transformer up-sample at the input dimension to recover. Universal Transformer (Dehghani et al., 2018) proposes a dynamic halting mechanism that determines the refinement steps for each token. Different from these works, BST utilizes attention information between question and token pairs and skims the input sequence at the block granularity accordingly.

**Efficient Transformer.** There are also many attempts for designing efficient Transformers (Zhou et al., 2020; Wu et al., 2019; Tay et al., 2020). Well studied model compression methods for Transformer models include pruning (Guo et al., 2020), quantization (Wang & Zhang, 2020), distillation (Sanh et al., 2019), weight sharing. Plenty of works and efforts focus on dedicated efficient attention mechanism considering its quadratic complexity of sequence length (Kitaev et al., 2019; Beltagy et al., 2020; Zaheer et al., 2020). BST is orthogonal to these techniques on the input dimension and therefore is compatible with them. We demonstrate this feasibility with the weight sharing model Albert (Lan et al., 2019) in Sec. 5.

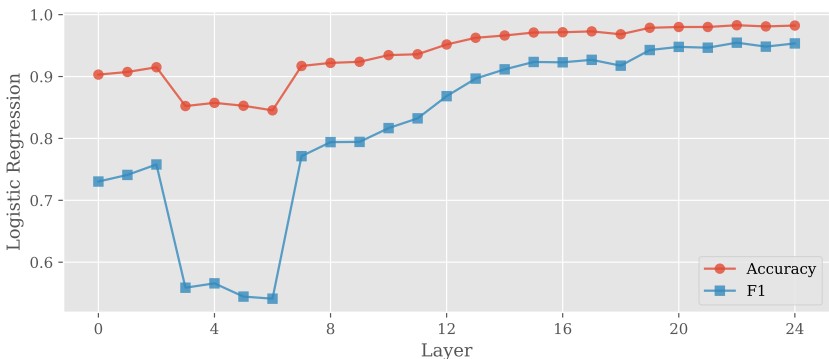

Figure 1: Accuracy and F1 score performance of logistic regression model predicting whether a block contains answer. The logistic regression model is built with attention values between question and target block as input feature.

## 3   PROBLEM FORMULATION: IS ATTENTION EFFECTIVE FOR SKIM

**Transformer.** Transformer model with multi-head self-attention mechanism calculates hidden states for each position as a weighted sum of input hidden states. The weight vector is calculated by parameterized linear projection query Q and key K as eq. 1. Given a sequence of input embeddings, the output contextual embedding is composed by the input sequence with different attention at each position.

$$Attention_{(Q,K)} = Softmax(\frac{QK^T}{\sqrt{d_k}}),  \tag{1}$$

where $Q, K$ are query and key matrix of input embeddings, $d_k$ is the length of a query or key vector. Multiple parallel groups of such attention weights, also referred to as attention heads, make it possible to attend to information at different positions.

QA is one of the ultimate downstream tasks in the NLP. Given a text document and a question about the context, the answer is a contiguous span of the text. To predict the start and end position of the input context given a question, the embedding of each certain token is processed for all transformer layers in the encoder model. In many end-to-end open domain QA systems, information retrieval is the advance procedure at coarse-grained passage or paragraph level. Under the characteristic of extractive QA problem that answer spans are contiguous, our question is that whether we can utilize such idea at fine-grained block granularity during the processing of transformer. Is the attention weights effective for distinguish the answer blocks?

To answer the above question, we build a simple logistic regression model with attention matrix from each layer to predict whether an input sentence block contains the answer. The attention matrices are profiled from a BERTlarge SQuAD QA model and reduced to block level following Eq. 2 (Clark et al., 2019). The attention from block [a,b] attending to block [c,d] is aggregated to one value. And the attention between a block and the question sentence, special tokens "[CLS]" and "[SEP]" are used to denote the attending relation of the block. Such 6-dimensional vector from all attention heads in the layer are concatenated as the final classification feature. The result is shown in Fig. 1 with attention matrices from different layers. Simple logistic regression with hand crafted feature from attention weight achieves quite promising classification accuracy. This suggests that the attending relationship between question and targets is indeed capable for figuring out answer position.

$$BlockAttention([a,b],[c,d]) = \frac{1}{b-a}\sum_{i=a}^{b}\sum_{j=c}^{d}Attention(i,j)  \tag{2}$$

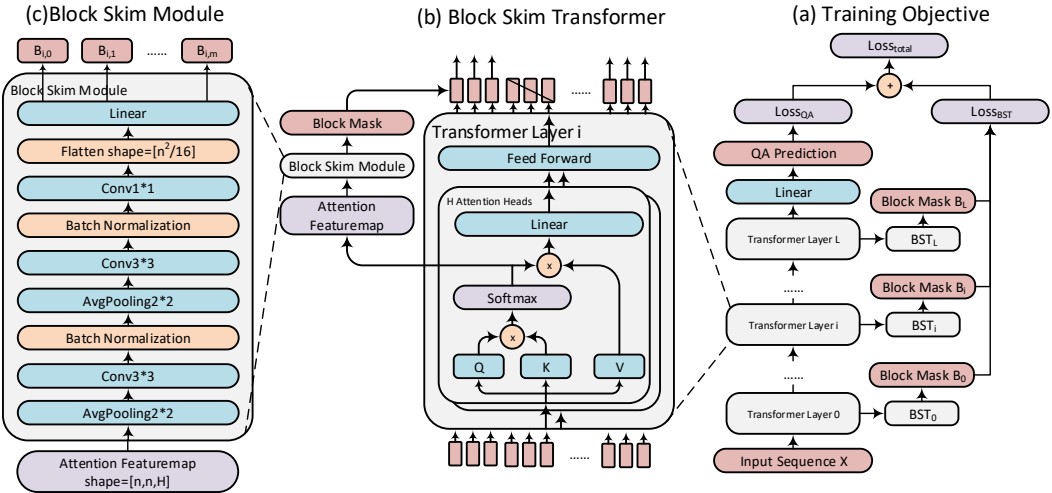

Figure 2: (a) The overview of Block Skimming Transformer. (b) A BST layer is composed of a Transformer layer and a Block Skimming Module. (C) The CNN-based BSM module details.

# 4 BLOCK SKIMMING TRANSFORMER (BST)

## 4.1 ARCHITECTURE OVERVIEW OF BST

We propose the Block Skimming Transformer (BST) model to accelerate the question answering task without degrading the answer accuracy. Unlike the conventional Transform-based model that uses all input tokens throughout the entire layers, our BST model accurately identifies the irrelevant contexts for the question in the early layers, and remove those irrelevant contexts in the following layers. As such, our model reduces the computation requirement and enables fast question answering.

In Sec. 3, we have shown that it is feasible to identify those tokens that are irrelevant to the question through a hand-crafted feature using the attentions relationship among tokens. However, using this approach could significantly hurt the question answering task accuracy as we show later. As such, we propose an end-to-end learnable feature extractor that captures the attention behavior better.

Fig. 2 shows the overall architecture of our BST model, where a layer is composed of a Transformer layer and a learnable Block Skim Module (BSM). The BSM adopts the convolutional neural network for feature extraction. The input is attention matrices of attention heads, which are treated as feature maps of multiple input channels. The output is a block-level mask that corresponds to the relevance of a block of input tokens to the question.

In each BSM module, we use convolution to collect local attending information and use pooling to reduce the size of feature maps. Two $3 \times 3$ convolution and one $1 \times 1$ convolution are connected with pooling operations intersected. For all the convolution operations, ReLU funcition (Hahnloser & Seung, 2001) is used as activation function. To locate the answer context blocks, we use a linear classification layer to calculate the score for each block. Also, two Batch Normalization layers (Ioffe & Szegedy, 2015) are inserted to improve the model accuracy.

Formally, we denote the input sequence of a transformer layer as $X = (x_0, x_1, \ldots, x_n)$. Then the attention matrices of this layer are denoted as $Attention(X)$. Given the attention output of a transformer layer, the $k_{th}$ block prediction result $B$ is represented as $B = BST(Attention(X))$, where BST is the proposed architecture. The main functions of BST is expressed as Eq. 3.

$$BST(Attention) = Linear(Conv_{1 \times 1}(Conv_{3 \times 3}(Pool(Conv_{3 \times 3}(Pool(Attention))))))$$ (3)

## 4.2 JOINT TRAINING OF QA AND BLOCK-SKIM CLASSIFIERS

There are two types of classifiers in our BST model, where the first is the original QA classifier at the last layer and the second is the block-level relevance classifier at each layer. We jointly train these classifiers so that the training objective is to minimize the sum of all classifiers' losses.

The loss function of each block-level classifier is calculated as the cross entropy loss against the ground truth label whether a block contains answer tokens or not. Equation 4 gives the formal definition. The total loss of the block-level classifier $\mathscr{L}_{BST}$ is the sum of all blocks that only contain passage tokens. The reason is that we only want to throw away blocks with irrelevant passage tokens instead of questions. Blocks that have question tokens or padding tokens are not used in the training process. To be more detailed, such blocks are pre-processed and dropped during the training process.

$$\mathscr{L}_{BST} = \sum_{b_i \in \{\texttt{passage blocks}\}} CELoss(b_i, y_i)$$
$$y_i = \begin{cases} 1 & \text{, block i has answer tokens} \\ 0 & \text{, block i has no answer tokens} \end{cases}$$

(4)

To calculate the final total loss $\mathscr{L}_{total}$, we introduce two hyper-parameters in Equation 5. We first use the hyper-parameter $\alpha$ so that different models and settings could adjust the ratio between the QA loss and block-level relevance classifier loss. We then use the other hyper-parameter $\beta$ to balance the loss from positive and negative relevance blocks because there are typically many more blocks that contain no answer tokens (negative bocks) than the blocks that do contain answer tokens (positive bocks). We explain how to tune those hyper-parameters for different models and settings later.

$$\mathscr{L}_{total} = \mathscr{L}_{QA} + \alpha \sum_{i_{th} \text{ layer}} (\beta \mathscr{L}_{BST}^{i,y=1} + \mathscr{L}_{BST}^{i,y=0})$$

(5)

Although we add the block-level relevance classification loss in the joint training, we do not actually throw away any blocks because it can skip answer blocks and the QA task training becomes unstable. In this sense, the block-level relevance classification loss can be viewed as a regularization method for the QA training as we force attention heads to better distinguish the answer blocks and non-answer blocks. As we show in the experiment, this regularization effect leads to accuracy improvement for the QA task.

## 4.3 USING BST FOR QA

We now describe how to use the BST model to accelerate the QA task. In the above joint training process, we add the BSM module in every layer. However, we only augment a specific layer with the BSM module during the inference to save computation and avoid heavy changes to the underlying Transformer model. As such, the layer index for augmenting is a hyper-parameter in our model.

Once the BSM-augmented layer is chosen, we split the input sequence by the block granularity, which is another hyper-parameter in our model. The model skips a set of blocks according to the BSM module results for the following layers. It should be noted that the BST training process does not throw away any blocks because if a relevant block with answer tokens is rejected, the training of the original QA task is confused and becomes unstable.

To maintain compatibility with the original Transformer model, we forward the skipped blocks directly to the last layer for the QA classifier. With those design features, BST works as an add-on component to the original Transformer model and is compatible with many Transformer variant models as well as model compression methods. In specific, we will demonstrate that BST works well with Transformer-based Roberta (Liu et al., 2019), which has a different pre-training objective and sequence encoding, and Albert (Lan et al., 2019), which shares weights among layers for a reduced model size.

We provide an analytical model to demonstrate the speedup potential of BST . Suppose that we insert the BSM module at the layer $l$ out of the total $L$ layers, and a portion of $k$ blocks remain for the following layers. The performance speedup is formulated by Equation 6 if we ignore the computation overhead in the BSM module. In fact, the computation of a single BSM module is much smaller than Transformer layers. For example, if $k = 1/3$ blocks remain after $l/L = 1/3$ layers, we achieve a $1.8\times$ ideal speedup. Similarly, if $k = 1/4$ blocks remain after $l/L = 1/4$ layers, the ideal speedup is $2.29\times$.

$$speedup = \frac{L \cdot N \cdot T_{layer}}{l \cdot N \cdot T_{layer} + (L-l) \cdot N \cdot k \cdot T_{layer}} = \frac{1}{1 - (1 - l/L)(1 - k)}$$

(6)

| | QA | | | | BST classifier | |
|---|---|---|---|---|---|---|
| | baseline | | BST | | layer 4 | middle layer |
| | EM | F1 | EM | F1 | F1 | F1 |
| Transformer-based models with BST on SQuAD | | | | | | |
| BERT$_{base}$ | 81.14 | 88.52 | 81.38 | 88.69 | 80.99 | 89.37 |
| BERT$_{large}$ | 86.53 | 92.82 | 87.12 | 93.20 | 82.48 | 93.32 |
| Roberta$_{base}$ | 82.02 | 89.65 | 82.40 | 89.91 | 81.88 | 90.35 |
| Roberta$_{large}$ | 86.03 | 92.81 | 86.17 | 92.95 | 78.93 | 86.56 |
| Albert$_{base}$ | 82.13 | 89.78 | 82.18 | 89.78 | 84.30 | 90.25 |
| Albert$_{large}$ | 84.26 | 91.51 | 84.71 | 91.73 | 78.93 | 93.02 |
| Avg. | 83.68 | 90.85 | 83.99 | 91.04 | 81.38 | 90.48 |
| BERT$_{base}$ BST on QA datasets | | | | | | |
| SQuAD | 81.14 | 88.52 | 81.38 | 88.69 | 80.99 | 89.37 |
| NewsQA | 51.45 | 66.57 | 52.28 | 67.43 | 53.74 | 67.02 |
| TriviaQA | 68.99 | 73.78 | 69.04 | 73.91 | 93.96 | 96.20 |
| HotpotQA | 58.76 | 75.56 | 59.55 | 75.93 | 69.67 | 75.36 |
| Natural Questions | 67.00 | 79.00 | 67.47 | 79.15 | 89.32 | 90.54 |
| Avg. | 65.47 | 76.69 | 65.94 | 77.02 | 77.54 | 83.70 |

Table 1: BST training results. (Upper) BST evaluated with different Transformer variant models and model sizes. (Lower) BST evaluated with BERT$_{base}$ on 5 QA datasets.

## 5 EXPERIMENT

### 5.1 EXPERIMENTAL SETUP

**Dataset.** We evaluate our method on 5 extractive QA datasets, including SQuAD 1.1 (Rajpurkar et al., 2016), Natural Questions (Kwiatkowski et al., 2019), TriviaQA (Joshi et al., 2017), NewsQA (Trischler et al., 2016) and HotpotQA (Yang et al., 2018). The diversity of these datasets such as various passage lengths and different document sources lets us evaluate the general applicability of the proposed BST method. We follow the setting of BERT model to use the structure of Transformer encoder and a linear classification layer for all the datasets.

**Model.** As we mentioned earlier, BST works as a plugin module to the oracle Transformer model, and therefore applicable to Transformer-based models. To illustrate this point, we apply our method to three different models including BERT, Roberta (Liu et al., 2019) with a different pre-training objective, and Albert (Lan et al., 2019) with parameter sharing layers. For all three models, we evaluate the base setting with 12 heads and 12 layers and the large setting with 24 layers and 16 heads as described in prior work (Devlin et al., 2018).

**Training Setting.** We implement the proposed method based on open-sourced library from Wolf et al. (2019). For each baseline model, we use the released pre-trained checkpoints [1]. We follow the training setting used by Devlin et al. (2018) and Liu et al. (2019) to perform the fine-tuning on the above extractive QA datasets. We initialize the learning rate to $3e - 5$ for BERT and Roberta and $5e - 5$ for Albert with a linear learning rate scheduler. For SQuAD dataset, we apply batch size 16 and maximum sequence length 384. And for the other datasets, we apply batch size 32 and maximum sequence length 512. We perform all the experiments reported with random seed 42. We train a baseline model and BST model with the same setting for two epochs and report accuracies from MRQA task benchmark for comparison. We use four V100 GPUs with 32 GB memory for training and report performance speedup on multiple different hardware platforms.

For the following experiments, we use the block size 32 unless explicitly mentioned. We set the hyper-parameter $\beta$ to 4 for all experiments and $\alpha$ to 1 except Albert. We use the $\alpha$ value of 0.05 for Albert. In the Albert model, the parameters of transformer layers are shared but BST modules

---

[1]We use pre-trained language model checkpoints released from `https://huggingface.co/models`

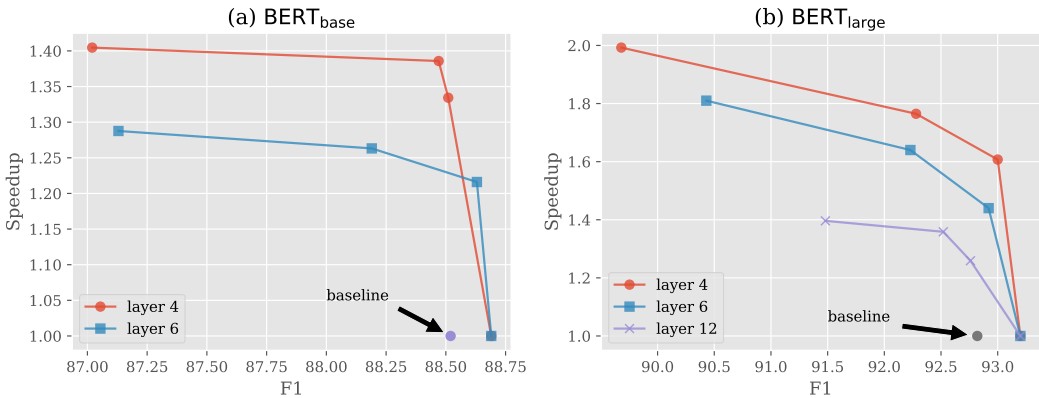

Figure 3: Speedup of BST on BERT$_{base}$ and BERT$_{large}$ augmented at different layers.

in our method do not share parameters. As such, we decrease the loss from BST to prevent model over-fitting and its impact on the QA task parameters.

**Performance Evaluation.** We measure the performance speedup of our method on the 2.20GHz Intel Xeon(R) Silver 4210 CPU with the batch size of one. On the GPU, the batch size of one could not fully utilize the GPU computation resource so the inference time is bottlenecked at the memory. Our evaluation scenario closely resembles prior work (Wu et al., 2019) that targets the mobile application domain. We evaluate BST with different layers and prediction thresholds of BST classifier to explore the trade-off between performance speedup and model accuracy. For example, a lower prediction threshold could lead to more skipped blocks, which means a better performance speedup. On the other hand, it also increases the chance of skipping answer blocks, which hurts the QA task accuracy.

### 5.2 BST as a Regularization Method

We first evaluate BST model as a regularization method to improve the QA task accuracy. Specifically, we compare the accuracy of three baseline models and their BST variants. In their BST versions, the BSM modules only participate the training process, and are removed in the inference task.

The upper half of Tbl. 1 shows the accuracy comparison on SQuAD dataset. By only changing the training process, BST improves the extractive QA accuracy for all baseline models. On average, BST exceeds the baseline by 0.32% in exact match and by 0.19% in F1 score. The accuracy improvement of BST is generally greater on large models. We attribute this to a stronger regularization effect for larger models. The results show the wide applicability of our method to different models.

Tbl. 1 also demonstrates the BST classifier F1 score trained jointly (but not used in this setting) with the baseline models. For simplicity, we only show the results of layer 4 and the middle layer (layer 6 for base and 12 for large model). On average, the block-level relevance classifier has a notably high F1 score even at early layers (averaged 81.38%) and even higher scores at the middle layer.

The lower half of Tbl. 1 shows BERT$_{large}$ results on multiple QA datasets. BST outperforms the baseline training objective on all datasets evaluated and exceeds with 0.52% exact match and 0.33% F1 score on average. The results show the wide applicability of our method to different datasets with varying difficulty and complexity. Meanwhile, we also observe a modest correlation between the block-level relevance classifier and the QA task. In other words, the BST classifier tends to be higher on datasets with a higher QA accuracy except for TriviaQA dataset.

### 5.3 QA Task Speedup with BST

We now demonstrate the BST's ability to accelerate the QA task. Fig. 3 demonstrates the performance speedup against F1 score evaluated with BERT$_{base}$ and BERT$_{large}$ model on SQuAD dataset. By tuning the prediction threshold of the BST classifier, we can trade-off between acceleration and accuracy loss. Here we evaluate the BST classifier with $0, 0.5, 0.9, 0.99$ prediction threshold with

| id | | Update Transformer | Skim Training | BSM | Augment Layer | Block Size | QA | | BST Classifier | |
|----|------|------|------|------|------|------|------|------|------|------|
| | | | | | | | EM | F1 | layer 4 | layer 6 |
| 1 | Baseline | ✓ | - | - | - | - | 81.14 | 88.52 | - | - |
| 2 | Vanilla BST | ✓ | | ✓ | All | 32 | 81.38 | 88.69 | 80.99 | 89.37 |
| 3 | Freeze Transformer | | | ✓ | All | 32 | 81.14 | 88.52 | 79.60 | 78.41 |
| 4 | Hand-crafted Feature | ✓ | | | All | 32 | 81.21 | 88.59 | 69.41 | 69.51 |
| 5 | Augment One Layer | ✓ | | ✓ | 4 | 32 | 81.20 | 88.63 | 77.26 | - |
| 6 | Skim Traning | ✓ | ✓ | ✓ | 4 | 32 | 79.27 | 86.83 | 84.76 | - |
| 7 | Block Size 1 | ✓ | | ✓ | All | 1 | 81.22 | 88.60 | 75.83 | 85.03 |
| 8 | Block Size 8 | ✓ | | ✓ | All | 8 | 81.25 | 88.63 | 76.12 | 87.04 |
| 9 | Block Size 16 | ✓ | | ✓ | All | 16 | 81.35 | 88.75 | 78.41 | 82.51 |
| 10 | Block Size 64 | ✓ | | ✓ | All | 64 | 81.39 | 88.65 | 87.77 | 90.76 |
| 11 | Block Size 128 | ✓ | | ✓ | All | 128 | 80.90 | 88.33 | 91.36 | 92.79 |

Table 2: Ablation studies of the BST design components of BERT$_{base}$ on SQuAD dataset.

classifier at layer 4 and the middle layer respectively. On BERT$_{base}$, BST achieves $1.38\times$ speedup with the same accuracy to the baseline. With a more aggressive skipping strategy, $1.4\times$ speedup is obtained with minor accuracy loss (less than 1.5%). On BERT$_{large}$, BST achieves $1.6\times$ speedup with minor accuracy improvement and nearly $1.8\times$ speedup with less than 0.5% F1 score degradation.

Generally, the specific layer for inserting the BSM module can be determined by hyper-parameter search according to Eq. 6. As shown in Fig. 3, skipping the irrelevant blocks at layer 4 tends to be better than that at the middle layer, which is layer 6 for BERT$_{base}$ and layer 12 for BERT$_{large}$. This is because more computation is reduced when skipping at earlier layers and the BST classifier already has a quite good prediction accuracy at early layers.

## 5.4 ABLATION STUDY

We compare our BST method with a series ablation of design components to study their individual effect. The experiments are performed based on the same setting as Sec. 5.1. We perform the experiments described in Tbl. 2, which has also the detailed results, and summarize the key finds as follows.

- (3) Instead of joint training as described in Sec. 4.2, we perform a two-step training. We first perform the fine-tuning for the QA task. We then perform the BSM module training with the baseline QA model frozen. In other words, we only use the BST objective and only update the weights in the BSM modules. Therefore, the QA accuracy remains the same as the baseline model, which is lower than the joint training (id 3). Meanwhile, the BST classifier also has a lower accuracy than the joint training especially at layer 6.

- (4) Instead of BSM module, we use the hand-crafted feature in Sec. 3. The resulted block-level relevance classification accuracy is considerably lower than our learned BST model.

- (5) Instead of adding BST module to all layers, we only deploy it into one layer. The experiment result shows that it is beneficial for the model to have BST loss added to every layer.

- (6) We skim blocks during the joint QA-BST training process. Because the mis-skimmed blocks may confuse the QA optimization, this training strategy results in considerable accuracy loss.

- (7-11) We evaluate the accuracy with different block sizes. Specifically, when the block size is 1, it is equivalent to skim at the token granularity. Our experimental result shows that the accuracy of BST classifier is better when the block size is larger. On the other hand, a larger block size also leads to less number of blocks and therefore the performance speedup becomes limited on the studied datasets. To this end, we choose the block size of 32 as a design sweet spot.

## 6 CONCLUSION

In this work, we provide a plug-and-play module BST to Transformer and its variants for efficient QA processing. Our empirical study shows that the attention mechanism in the form of a weighted feature map can provide instructive information for locating the answer span. In fact, we find that an attention

map can distinguish between answers and other tokens. Leveraging this insight, we propose to learn the attention in a supervised manner. In effect, BST terminates irrelevant blocks at early layers, significantly reducing the computations. Besides, the proposed BST training objective provides attention mechanism with extra learning signal and improves QA accuracy on all datasets and models we evaluated. With the use of BST module, such distinction is strengthened in a supervised fashion. This idea may be also applicable to other tasks and architectures.

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
