# OpenReview forum: "Block Skim Transformer for Efficient Question Answering"
_ICLR.cc/2021/Conference — Reject_

### Official Review · AnonReviewer2 · 2020-10-17
**Cool idea but the model is confusing and results are not convincing**

**Rating:** 5
**Confidence:** 4

**Review:**


Summary: This paper presents the "Block Skim Transformer" for extractive question answering tasks. The key idea in this model is using a classifier, on the self-attention distributions of a particular layer, to classify whether a large spans of non-contiguous text (blocks) contain the answer. If a block is rejected by the classifier, it is excluded in subsequent layers of self-attention. During training, no blocks are thrown away and the classifier is applied to every layer to provide a regularization effect, which leads to small improvements in performance in 5 datasets. During inference, blocks are thrown away at a fixed layer. The reduction in sequence length leads to ~1.5x batch size 1 speed improvements.

-----------------------------------

Strengths of the Paper

1. Lots of experiments, using 5 different datasets and many different pretrained LMs like BERT, RoBERTa and ALBERT. Small but consistent improvements in most settings. A detailed ablation study testing hyperparameters and different components of the model.

3. Cool idea of dropping tokens entirely, this is relatively unexplored in the space of efficient transformer models.

-----------------------------------

Weaknesses of the Paper

1. Training-Inference mismatch: It seems to me the proposed model has a strong mismatch during training and inference. During training, none of the blocks are dropped. During inference, a number of tokens are dropped at a particular layer (which is a inference time hyperparameter). It seems counter-intuitive to me that the subsequent layers of the model are able to successfully process a partial input, especially when this was not done during training. Also, this line is confusing "we forward the skipped blocks directly to the last layer for the QA classifier." Isn't this another training-inference mismatch? During training the QA classifier got the final layer representations for every block, not the intermediate layer representation.

2. Improvements are quite small: in terms of QA performance, the improvement is just 0.3 F1 on an average across datasets. Is this improvement statistically significant? Even in terms of speed, a 1.6x batch size 1 improvement seems relatively modest. Are the speed / accuracy improvements better on tasks requiring long input sequences? Intuitively, the Skim idea seems more well-suited to tasks having inputs much longer than 384 tokens, perhaps you could use some of the benchmarks in the efficient transformer papers?

3. How does this method compare to the numerous efficient transformers that have been proposed (in terms of attention, or adaptive layers)? While I don't expect comparisons against all of them, atleast 1-2 comparisons should be done to ground the observations better. Modifying attention distributions to be sparse over tokens seems like a more general way of getting the same effect as the block skim idea, and this also avoids the training-inference mismatch.

-----------------------------------

Overall Recommendation

The idea of dropping tokens is interesting, but I'm not very convinced by the results. Performance improvements are modest and I'm really confused about the train-inference mismatch in the proposed architecture. More comparisons with a few alternative efficient transformer papers are needed to make this a stronger submission. Overall I'm leaning reject, but open to increasing my score slightly if I get more clarification regarding weakness #1.

-----------------------------------

Minor: I don't really understand Eq 2. What's the final vector that's used for logistic regression? How exactly is the aggregation done between the question and a block of text?

----------------------------------

**After Author Response**

Thank you for the detailed clarifications. I've raised my score to 5 since I found the random block skipping experiment quite interesting and surprising. However, weakness #2 and #3 are fairly important in my opinion and I don't think the response sufficiently addresses those weaknesses. I encourage the authors to show their method works on longer sequence datasets (where skimming intuitively makes sense) and compare against 1-2 efficient transformers using sparse attention, for better grounding the improvements in existing literature.

---

> ### Author Response · Authors · 2020-11-19
> **Response to reviewer 2**
>
> Thank you very much for the comments.
> We provide clarifications to your questions and comments as follows.
>
> **1. Question: Training-inference mismatch.**
>
> Answer:
>
> We do use slightly different paradigms for training and inference phases. However, our results indicate that its impact is negligible: the accuracy of our model is comparable or even better than the baseline models without such a mismatch. We have designed an empirical experiment to measure the impact of the mismatch on the inference accuracy raised by the reviewer. We randomly skim blocks without correct answers at different layers with a native QA model on SQuAD. The averaged accuracy degradation of all layers is 0.258%. This strongly indicates that the mismatch is negligible for the inference. In fact, this “training-inference” mismatch also exists in other NLP tasks such as text summarization or machine translation, where we train with the next token but generate a long sequence.
>
> On the other side, our experiments demonstrate that “fixing” this mismatch, meaning that we actually drop the skimmed blocks during the training, leads to accuracy degradation (refer to ID.6 of Tbl. 2). The reason is that answer blocks may be wrongly skimmed before the block-skimming module is fully trained. Given this reason, we plan to investigate a two-stage training process to eliminate this “train-inference” mismatch. We first jointly train the baseline model and block-skimming module to a near convergence point without dropping blocks. We then start block-dropping training.
>
> Another reason is that we design our model to be an add-on plugin to the native oracle model. That is we profile attention information from the attention layer and training the BST blocks, which makes our method compatible to different models structure and compression methods.
>
>
>
> **2. Question: Improvements are quite small.**
>
> Answer:
>
> We have appended multiple runs of our BST method to demonstrate that its improvement is statistically significant. Specifically, we have run several experiments 5 times given the time constraints. The average accuracy is 81.41 with a small 0.27 standard deviation and speedup is 1.6X (compared with baseline accuracy of 81.17). We also observe that with identical training settings (such as the random seed), our method with BST regularization consistently outperforms baseline models. We will add full results in the next version if accepted.
>
> The reviewer is correct regarding that the benefits of our method are more significant for longer sequences. Thank you for the advice and we plan to evaluate the longer dataset used in the other papers.
>
> **3. Question: How does this method compare to the numerous efficient transformers that have been proposed (in terms of attention, or adaptive layers)?**
>
> Answer:
>
> Our method is generally orthogonal and applicable to other model efficiencies optimization methods such as pruning, quantization, and distillation, as we have demonstrated with AlBert experiments. Several recent efficient attention mechanisms (the X-former works) optimize the quadric attention computation by adding constraints on the attention positions such as the passage-wide attention in the DeFormer. Those mechanisms still need to compute all token embeddings even though the attention is sparse. As such, our method that skims tokens is complementary to those works.
>
> **4. Question: I don't really understand Eq 2.**
>
> Answer:
>
> We revised the writing and explanation of Eq.2 which was hard to understand. We manually design the feature that measures the relevance between the question and each passage block for an empirical study in sec.3. It is concatenated by block attention between each block and question sentence, special tokens “[cls]” and “[sep]”. Such 6-dimensional feature from all attention heads is combined as the final feature vector for classification.

---

> > ### Comment · AnonReviewer2 · 2020-11-21
> > **Thanks for the detailed replies, one more clarification would be great!**
> >
> > Thanks for the detailed replies! Could you also clarify this part of my weakness #1?
> >
> > > Also, this line is confusing "we forward the skipped blocks directly to the last layer for the QA classifier." Isn't this another training-inference mismatch? During training the QA classifier got the final layer representations for every block, not the intermediate layer representation.

---

> > > ### Author Response · Authors · 2020-11-23
> > > **Response to reviewer 2**
> > >
> > > Thank you for the prompt reply.
> > >
> > > We would like to clarify that the reason why we forward the embeddings of skimmed blocks to the last layer is to simplify the processing of the QA task. In specific, the number of tokens that our method generates is identical to the number of tokens in the baseline model. As such, we can directly use the QA classifier without any modification. If we were to skim blocks without forwarding, we need to post-process the results from the QA classifier because the indices of the passage after skimming blocks are different from the original passage.
> > >
> > > We have designed an extra experiment and shown that directly skimming tokens and forwarding skimmed tokens both have a negligible impact for the QA accuracy. In fact, directly skimming tokens has a slightly smaller impact than forwarding skimmed tokens. For this experiment, we use the unmodified BERT model on SQuAD dataset. We randomly skim blocks that do not have answers to the question. We compare the final QA accuracy with two cases: directly skimming blocks and forwarding skimmed blocks. The former leads to a 0.204% accuracy drop and the latter causes a 0.258%. However, in the former case, we need to use the block skimming mask to locate the final answer from the text. These results show that this mismatch is negligible for the inference. We will revise our manuscript to explain these problems and add the extra experiments as ablation studies in section 5.

---

### Official Review · AnonReviewer3 · 2020-10-29
**Well-written paper with comprehensive experiments & impressive speedup, however, the model requires pre-defined supervision for tokens for skimming**

**Rating:** 6
**Confidence:** 4

**Review:**

This paper introduces Block Skim Transformer (BST), a variant of Transformer that skips tokens for self-attention so that it can pay less attention to less import tokens and achieve speedup. Although previous work studied the skimming mechanism for RNNs, this is the first paper that uses such an idea for transformers.

The model is pretty straight-forward - a block skim module in each transformer layer has a token-wise classifier that identifies which token to skip. One thing to note is that this model is specific to question answering (as the title indicates), because the supervision for tokens to skip and not to skip are pre-determined by identifying whether the token is part of the answer or not.

Strengths of the paper:
1) The problem is well-motivated with extensive discussion of related work.
2) The model is straight-forward and the description is easy to follow.
3) The paper includes extensive experiments using six different types of base pretrained models and five QA datasets, and comprehensive ablations.
4) The model achieves impressive speedup, e.g., without more than 0.5% performance degradation, the model achieves x1.4 speedup and x1.8 speedup with BERT base and BERT large, respectively.

Weakness of the paper:
The most important concern I have is that the model requires pre-defined supervision for tokens in order to train BST module. For instance, this paper trains the model to skim all the tokens that are not the answer token (which is the reason that it is specific to question answering). This means that BST module is trained for an end task (finding the answer tokens), rather than being trained to identify less important tokens for the end task. This is also a main difference from previous work that incorporates skimming modules for RNN, as most of them treat blocks to skip as latent variables and hence are easy to generalize to any tasks. This makes the model significantly less novel, and this fact was not apparent in Section 1 and 2, before reading the actual description of the model.

---

> ### Author Response · Authors · 2020-11-19
> **Response to reviewer 3**
>
> Thank you for your valuable comments and advice, here are some responses about your concerns.
>
> Question:
> Concerns about generality to end tasks compared with previous skimming works.
>
> Answer:
>
> As a matter of fact, we train the BST module to identify whether the oracle transformer model finds the answer. The key insight is that attention to the answer sentence or block is distinguishable. And as shown in sec.3, attention is indeed capable to do so. Our motivation is to utilize (and also improve) the information of self-attention relation, which is not explored in the previous skimming modules. We show that the attention mechanism has a distinguishable behavior between answer tokens and others in section 3. We believe this is our novelty compared to many efficient transformer works optimizing the quadric attention mechanism. We concede that the supervision of target blocks does become a limit for our method to be extended to other tasks.  We are planning to use other approaches to make the model discover the key blocks directly through the objective of the task. This remains to be our future work since that is out of the scope of this paper.

---

### Official Review · AnonReviewer1 · 2020-10-29
**Nice idea for improving QA efficiency by skimming some blocks.**

**Rating:** 6
**Confidence:** 4

**Review:**

This paper introduces a block skimming approach for QA.

The main idea is to use a light-weight module that uses the attention weights to predict which blocks of input are not useful for answering a question and use this information to avoid processing these blocks in further layers. Empirical evaluation shows that this approach can reduce computation and provide as much as 1.6X speedup with no drop in accuracy and even further speedups for more small drops in accuracy. Furthermore, using the block skimming component i.e. recognizing which blocks are useful works as a regularizer and can even increase QA performance by modest amounts.

The main strengths of the work are:

1. The main idea is interesting and well-motivated. We want good solutions for getting large models to run efficiently.
2. The evaluation on multiple datasets and across three types of transformer based models show that skimming can provide modest but consistent gains and can provide speedups.

The main weaknesses are:

1. Efficiency is a key point of this method. Yet, there are no external efficiency related baselines compared (e.g. DistillBERT). I buy the argument that this method can be seen as orthogonal to the other methods but it will be useful at the very least to show how the speed-ups fare when compared to other methods. The only comparison we have is with Albert but we don’t have any speed-up comparisons with it either.
2. The gains with using skimming as a regularization method is limited.


Other Questions and Suggestions:


1. Given the limited improvements in effectiveness, it will be worth reporting averages of a few runs (three-five) for at least  some of the rows in Table 1.

2. It will be useful to know the speed-ups for some of the block sizes. Is it possible to add these numbers in Table 2?

3. DeFormer [1], a paper that appeared in ACL 2020, is a related work that simply removes attention computation across question and passage blocks in lower layers. This idea here is more general than the DeFormer work but is close enough to warrant a discussion in the paper at the very least.

4. Can you provide some intuitions for the specific design choices for the Block Skimming Module?

5. It will be interesting to show if you are able to run BERT-large in the skim mode faster than  BERT-base in the original model. This will highlight the fact that skimming method can be used to improve

6. What kinds of blocks get ignored by the model? A post-hoc manual analysis would have been nice.

---

> ### Author Response · Authors · 2020-11-19
> **Response to reviewer 1**
>
> Thank you for your precious advice. The following are some comments.
>
> **1. Question: Inference experiments are motivated by a "mobile" use case setting, where a CPU is used with batch size one.**
>
> Answer:
>
> There are two reasons why we choose the CPU for the speedup evaluation. The first is that the CPU closely represents the platform for the practical inference scenario. For example, Facebook publicly acknowledges that it uses GPU for training and CPUs for inference in its datacenter. On the other hand, modern GPUs are already powerful so that the inference with a batch size one cannot fully occupy the GPU. As a consequence, even though our method skims a significant number of blocks without accuracy degradation, the reduced computation cannot translate to a meaningful speedup.**:
>
>
>
> **2. Question: Given the limited improvements in effectiveness, it will be worth reporting averages of a few runs (three-five) for at least some of the rows in Table 1.**
>
> Answer:
>
> We have run several experiments 5 times given the time constraints. The average accuracy is 81.41 with a small 0.27 standard deviation and speedup is 1.6X (compared with baseline accuracy of 81.17). We also observe that with identical training settings (such as the random seed), our method with BST regularization consistently outperforms baseline models. We will add these results in the next version if accepted.
>
> **3. Qustion: It will be useful to know the speed-ups for some of the block sizes.**
>
>
>
> Answer:
>
> We measure the speedup of different batch sizes and append it to Tbl. 2. A larger batch size will lead to less speedup because fewer tokens are skimmed. However, the skimming classifier has better accuracy with a proper batch size selected. So we have tuned this hyper-parameter and choose it to be 32.
>
> **4. Question: This idea here is more general than the DeFormer work but is close enough to warrant a discussion in the paper at the very least.**
>
> Answer:
>
> We will add a discussion to Deformer in the next version of our paper. Deformer aims to reduce the computation of the attention mechanism by limiting the attention weights at lower layers to question-wide and passage-wide self-attentions (i.e., no attention between question and passage). This is orthogonal to our work that focuses on removing the hidden states at higher layers by utilizing the lower layer attention. In other words, we can apply our method BST on DeFormer as well.
>
> On the other hand, applying our method on DeFormer would bring an interesting question on which layer to perform the skimming task. One of the important insights in our work is that the attention weights at the lower layers have a strong correlation between question and answer (Fig.1 in Sec.3). This insight can explain why DeFormer hurts the QA accuracy by removing the question-passage attention at the lower layers.
>
> **5. Question: Can you provide some intuitions for the specific design choices for the Block Skimming Module?**
>
> Answer:
> We choose the convolutional neural network based architecture for the block skimming module because it is trainable and weight-efficient. Besides convnets, we have also tried other ways such as aggregating the attention features, which all degrade the accuracy. We also find the skimming accuracy is relatively insensitive to the convolution parameters such as filter size, number of layers, and choices of pooling function.
>
>
> **6. Question: It will be interesting to show if you are able to run BERT-large in the skim mode faster than BERT-base in the original model.**
>
> Answer:
>
> Our method could greatly improve the accuracy-latency space of the attention-based models. In particular, we can accelerate the latency of BERT-large by 2x. However, it is still slower than the original BERT-base because the latency of the original BERT-large is 3x of BERT-base. Fortunately, our method is orthogonal to other efficient Transformer models.

---

> > ### Author Response · Authors · 2020-11-19
> > **Continued: Response to reviewer 1**
> >
> > **7. Question: What kinds of blocks get ignored by the model? A post-hoc manual analysis would have been nice.**
> >
> > Answer:
> >
> > As a manual post-hoc analysis, we show an example from the bert-base model on SQuAD dataset with BSM at layer 4. We use “|” to identify the block boundaries with a granularity of 32 tokens. Blocks with a strikethrough are skimmed and the correct answers are underlined. In this sample, 8 blocks are skimmed off of the total 12 blocks. Except for the very first block with the question part, three paragraph blocks are skimmed. In this example, the answer is exactly located on the boundary of blocks and two adjacent blocks are kept. We hypothesize this is because of the locality in the attention and convolution net we used.
> >
> > > [CLS] who played quarterback for the broncos after peyton manning was benched? [SEP] following their loss in the divisional round of the previous season's playoffs, the denver | ~~broncos underwent numerous coaching changes, including a mutual parting with head coach john fox ( who had won four divisional championships in his four years as broncos head coach ),~~ | and the hiring of gary kubiak as the new head coach. under kubiak, the broncos planned to install a run - oriented offense with zone blocking | ~~to blend in with quarterback peyton manning's shotgun passing skills, but struggled with numerous changes and injuries to the offensive line, as well as manning having his worst~~ | ~~statistical season since his rookie year with the indianapolis colts in 1998, due to a plantar fasciitis injury in his heel that he had suffered since the~~ | ~~summer, and the simple fact that manning was getting old, as he turned 39 in the 2015 off - season. although the team had a 7 – 0 start | , manning led the nfl in interceptions. in week 10, manning suffered a partial tear of the plantar fasciitis in his left foot. he set~~ | the nfl's all - time record for career passing yards in this game, but was benched after throwing four interceptions in favor of backup quarterback brock osweiler | , who took over as the starter for most of the remainder of the regular season. osweiler was injured, however, leading to manning's return during the | ~~week 17 regular season finale, where the broncos were losing 13 – 7 against the 4 – 11 san diego chargers, resulting in manning re - claiming the starting quarterback~~ | ~~position for the playoffs by leading the team to a key 27 – 20 win that enabled the team to clinch the number one overall afc seed. under defensive coordinator wade~~ | ~~phillips, the broncos'defense ranked number one in total yards allowed, passing yards allowed and sacks, and like the previous three seasons, the team has continued [SEP]~~

---

### Official Review · AnonReviewer4 · 2020-10-29
**Interesting idea but unconvincing execution**

**Rating:** 4
**Confidence:** 4

**Review:**

This paper proposes Block Skim Transformer (BST), a variation of Transformer-based architectures for QA where a newly proposed module predicts the relevance of each 'block' of context text and masks less relevant blocks out. The authors show 1.6x speedup during inference across 5 QA datasets, with a small increase in accuracy.

While I find the motivation and the high level idea of using Attention weights to decide block relevance interesting, there are too many gaps in both the idea and the writeup at this point.  Below I list a few of them:

* The way the block relevance part of the model is trained, it should really only work for factoid or single-hop QA tasks. This is because a block is marked relevant only if it contains answer words. On the other hand, in multi-hop or mulit-fact QA tasks (which is where the focus on the QA community currently is), this simple relevance criteria just doesn't work. E.g., in a question like, "Where was the 44th president of the USA born?", any information about WHO the 44th president is would not be marked as relevant, yet it's critical to answer the question.  The same applies to so-called "comparison" questions in HotpotQA.

* Many important details are missing, which makes it difficult to assess the setup and the value of the findings. E.g., HotpotQA's so-called distractor setting (which is the easier one) has 10 paragraphs as context. How are these even fed into your baseline and BST models, when you have a max token limit of 512? Similarly for TriviaQA, where there are many (six, I think) evidence documents from Wikipedia.

* The technique, by design, needs all context in the bottom 1/3 or 1/4 layers of the Transformer architecture.  This means it cannot handle any longer context than the baseline; everything must fit into the 512 token limit, which is arguably the biggest bottleneck of most of the current Transformer architectures. In other words, the paper doesn't address this problem, which is more important than obtaining a 1.6x speedup.

* Eqn (2) is confusing: what are a and b? Start and end of a block? If so, this computation is the sum of all self-attention within the block from a to b. The text mentions using the attention between the question and the block for assessing relevance. The precise equation for that cross-attention would be more valuable here.

* Eqn (6) is confusing: I was expecting a quadratic term, like N^2 in the numerator and k^2 N^2 in the right hand side of the denominator. What is "Layer" in this equation?  Also, please check your simplification, it doesn't seem correct (note, e.g., that as k increases, your speedup expression on the right also increases, but it should decrease). The correct term seems something like L / (l + (L-l) k^2)  =  L / (k^2 L + (1-k^2) l).

* The speedup of 1.6x, while valuable, doesn't seem like a make-or-break decision when deciding to use a Transformer model or not at inference time.  I was hoping for a larger gain when keeping only a fraction of the context.

* What about training time -- is there any speed up there?  Do I understand correctly the answer is "no" because, I think, you keep all blocks during training time in case the relevance decision is incorrect?

* Inference experiments are motivated by a "mobile" use case setting, where a CPU is used with batch size one.  Is it correct to assume there isn't any significant speedup on a GPU with a larger batch, e.g., when computing Dev/Test numbers for the 5 QA datasets considered?

The writing has many typos and small mistakes, and would benefit from through proof reading.

A convincing clarification of the above concerns could change my mind, but at this point I don't think the paper is ready for publication.

---

> ### Author Response · Authors · 2020-11-19
> **Response to reviewer 4**
>
> Thank you for your review and precious comments.
> We would like to append some clarification here to explain some misunderstanding.
> In this work, we explore a plug and play method upon Transformer QA models for acceleration.
> Instead of directly solving problems of the QA task itself, our idea is to accelerate existing QA models, which is hopefully applicable to many attention based methods.
>
> **1. Question: It should really only work for factoid or single-hop QA tasks.**
>
> Answer:
>
> We believe our method also works for the multi-hop/multi-fact tasks. In fact, we have performed experiments on the HotpotQA dataset and the result shows that accuracy degradation caused by skimming is similar or even smaller than other single-hop QA datasets. On the HotpotQA dataset, our bert-base baseline model has an accuracy of 58.76. The BST model has an accuracy of 59.55 as a pure regularization method and an accuracy of 58.79 when actually skimming blocks.
>
> We agree with the reviewer that skimming evidence in the plain text would hurt the accuracy dramatically on the multi-fact HotpotQA dataset. We suggest two reasons why our method works fine on such datasets.
> 1. Instead of skimming plain text tokens, our method skims the contextual embeddings because the block skimming module is located in the middle of Transformer layers. The skimmed embeddings have already been processed by several transformer layers and therefore contain the context information.
> 2. Our block skimming module predicts whether a block contains the final answer with the information from attention behavior. If the skimming module reaches a high accuracy, it means that the baseline model has already located the final answer. As such, skimming blocks without answers would not hurt the QA accuracy. In fact, on a dataset like HotpotQA, the skimming module achieves a high accuracy at a higher layer than the simpler datasets.
>
> We would like to point out that although the QA accuracy on the HotpotQA dataset  is indeed considerably lower than other single-hop datasets, this is the problem of the baseline model, and hence out of the scope of our work.
>
> **2. Question: Many important details are missing, which makes it difficult to assess the setup and the value of the findings.**
>
> Answer:
>
> We have added more details that reviewer requests. When training with the max sequence limit of 512, we split the input sequence into multiple sliding spans with window size of 128. This is the default setting used by the vanilla BERT model, which we do not change.
>
> For the experiments in Sec. 5, we also follow the training setup used by its vanilla model. This highlights the feature of our method, which can work as an add-on component without any changes to the vanilla model. As we have explained in Sec 5.4 with ablation case No.3, we can freeze the trained QA model and only train our BST components. In such a case, there is no impact on the original QA model training process. We add the block skimming loss to the total training loss because our results show that its added regularization effect improves both the QA and block skimming accuracy.
>
> **3. Question: Eqn (2) is confusing.**
>
> Answer:
>
> We have added clarifications to the equation. Eqn. (2) computes the block attention from block with boundary [a,b] to another block [c,d]. The cross attention between a block [a,b] and question [c,d] is the summed attention as Eqn.(2). We use this equation to explain our insight that attention is informative enough for skimming.
>
> $BlockAttention([a,b],[c,d])=\frac{1}{b-a}\sum_{i=a}^b{\sum_{j=c}^d{Attention(i,j)}}$
>
> **4. Question: Eqn (6) is confusing.**
>
> Answer:
>
> We use this equation to estimate the ideal speedup of the input reduction idea.
> One of our assumptions is that the computation time or complexity is linearly proportional to the input sequence length, which holds true for all operations in a transformer layer except the self-attention calculation. Without any loss of generality, we make this linear assumption as a conservative approximation.
> We use $T_{layer}$ to denote the processing time of a layer.
> And the input sequence length is N.
> Suppose we skim a proportion of k blocks at layer l of total L layers.
> The equation is shown as following.
> We will revise this equation and explain our assumptions better.
>
> $speedup = \frac{LNT_{layer}}{lNT_{layer}+(L-l)kNT_{layer}} = \frac{1}{1-(1-l/L)(1-k)}$
>
> **5. Question: What about training time -- is there any speed up there?**
>
> Answer:
>
> Our work targets the inference time acceleration and does not reduce the training time.
> According to our experiments (ablation case NO. 6 in Sec 5.4), skimming blocks during the training significantly hurt the QA accuracy. As such, our method adds extra overhead to the training process. Reviewer 2 raised concern over this “training-inference mismatch”, please also refer to our response to this concern (reviewer 2, question 1).

---

> > ### Author Response · Authors · 2020-11-19
> > **Continued: Response to reviewer 4**
> >
> > **6. Question: Inference experiments are motivated by a "mobile" use case setting, where a CPU is used with batch size one.**
> >
> >
> > Answer:
> >
> > There are two reasons why we choose the CPU for the speedup evaluation. The first is that the CPU closely represents the platform for the practical inference scenario. For example, Facebook publicly acknowledges that it uses GPU for training and CPUs for inference in its datacenter. On the other hand, modern GPUs are already powerful so that the inference with a batch size one cannot fully occupy the GPU. As a consequence, even though our method skims a significant number of blocks without accuracy degradation, the reduced computation cannot translate to a meaningful speedup.

---

### Decision · Program_Chairs · 2021-01-07
**Final Decision**

**Decision:**

Reject

**Comment:**

The paper presents a model for question answering where blocks of text can be skipped and only relevant blocks are further processed for extracting the answer span.
 The reviewers mostly praised the general idea.
 R3 raised concerns on generalizability of the presented approach.
R4 raised several issues regarding presentation and clarity.
R2 and R4 have concerns regarding execution and find some of the results unconvincing.
While I don't necessarily share R2s concern on small improvements (improvements are still statistically significance), and despite the approach being very interesting, there are several issues that reviewers pointed out and wasn't resolved after discussions.